# Detection of Immune Escape and Basal Core Promoter/Precore Gene Mutations in Hepatitis B Virus Isolated from Asymptomatic Hospital Attendees in Two Southwestern States in Nigeria

**DOI:** 10.3390/v15112188

**Published:** 2023-10-31

**Authors:** Oguntope Adeorike Sobajo, Judith Uche Oguzie, Benjamin Adegboyega, Philomena Eromon, Christian Happi, Isaac Komolafe, Onikepe Folarin

**Affiliations:** 1Department of Biological Sciences, Faculty of Natural Sciences, Redeemer’s University, Ede 232102, Osun State, Nigeria or sobajooa@abuad.edu.ng (O.A.S.); oguziej@run.edu.ng (J.U.O.); happic@run.edu.ng (C.H.); toshokomolafe@gmail.com (I.K.); 2African Centre of Excellence for Genomics of Infectious Diseases, Redeemer’s University, Ede 232102, Osun State, Nigeria; adegboyegabb@gmail.com (B.A.); eromonp@run.edu.ng (P.E.); 3Department of Biological Science, College of Sciences, Afe Babalola University, Ado-Ekiti 360101, Ekiti State, Nigeria

**Keywords:** HBV genotype E, immune escape mutations, major hydrophilic region, basal core promoter/precore region, southwest, Nigeria

## Abstract

Several mutations in the surface (S), basal core promoter (BCP), and precore (PC) genes of the hepatitis B virus have been linked to inaccurate diagnosis and the development of immune escape mutants (IEMs) of the infection, which can lead to chronic infection. Understanding the prevalence and spread of these mutations is critical in the global effort to eliminate HBV. Blood samples were collected from 410 people in Osun and Ekiti states, southwest Nigeria, between 2019 and 2021. Participants were drawn from a group of asymptomatic people who were either blood donors, outpatients, or antenatal patients with no record of HBV infection at the medical outpatients’ unit of the hospital. DNA was extracted from plasma using a Qiagen DNEasy kit, followed by nested PCR targeting HBV S and BCP/PC genes. The Sanger sequencing method was used to sequence the positive PCR amplicons, which were further analyzed for IEMs, BCP, and PC mutations. HBV-DNA was detected in 12.4% (51/410) of individuals. After DNA amplification and purification, 47.1% (24) of the S gene and 76.5% (39) of the BCP/PC gene amplicons were successfully sequenced. Phylogenetic analysis showed that all the HBV sequences obtained in this study were classified as HBV genotype E. Mutational analysis of the major hydrophilic region (MHR) and a-determinant domain of S gene sequences revealed the presence of three immune escape mutations: two samples harbored a T116N substitution, six samples had heterogenous D144A/N/S/H substitution, and one sample had a G145E substitution, respectively. The BCP/PC region analysis revealed a preponderance of major BCP mutants, with the prevalence of BCP double substitutions ranging from 38.5% (A1762T) to 43.6% (G1764A). Previously reported classical PC mutant variants were observed in high proportion, including G1896A (33.3%) and G1899A (12.8%) mutations. This study confirms the strong presence of HBV genotype E in Nigeria, the ongoing circulation of HBV IEMs, and a high prevalence of BCP/PC mutants in the cohorts. This has implications for diagnosis and vaccine efficacy for efficient management and control of HBV in the country.

## 1. Introduction

The hepatitis B virus infection accounts for one-third of all liver cancer deaths globally, with hepatocellular carcinoma (HCC) being the leading cause of death due to cancer [1,2]. In low- and middle-income countries, hepatitis B-related HCC accounts for two-thirds of all liver cancer cases [3]. According to the World Health Organization (2019) estimate, more than 820,000 HBV-related deaths occur yearly due to liver cirrhosis and HCC, while 296 million individuals worldwide suffer from chronic hepatitis B. About 20 million Nigerians are estimated to be infected with HBV [4,5].

HBV is a partly double-stranded virus that has four overlapping open reading frames (ORFs) that code for the surface proteins (PreS1/PreS2/S): the polymerase (Pol), the capsid proteins (PreC/C), and the X protein [6]. The HBV S gene encoding the small HBsAg contains clusters of B-cell epitopes known as the major hydrophilic region (MHR), spanning amino acid position 99–169 [6]. Unlike other DNA viruses, HBV has a variable genome with ten genotypes and sub-genotypes based on 4–8% intra-genotypic divergence [7]. The HBV genotype has a particular geographic distribution worldwide, with genotypes A, D, and E being the most prevalent in Africa. The most prevalent HBV genotype in West Africa is E [8]. HBV has a higher mutation rate than other DNA viruses because it replicates via an intermediate RNA and a reverse transcriptase incapable of proofreading [9].

Despite chronically increasing HBV DNA levels and active liver illness, HBV strains from various nations have been shown to fail to produce HBeAg. This failure has been linked to mutations in the basal core promoter (BCP) and precore (PC) regions [10,11,12]. Specifically, it has been noted that G1896A and G1899A result in a translational stop codon that is predicted to affect HBeAg expression, ultimately leading to HBeAg loss and sometimes HBeAg-negative chronic HBV (CHB) infection [13,14,15]. According to reports, 1762T/1764A double BCP mutations have been suggested to be one of the risk factors for HCC [16,17,18,19]. Furthermore, these mutations may lead to long-term HBV therapy, making patients susceptible to nucleos(t)ide analog drug resistance gene evolution [20,21,22]. With an estimated incidence ranging between 2.5% and 40% over the entire country, Nigeria has long been one of the most HBV-endemic nations in sub-Saharan Africa [23,24]. Despite this, there is a severe shortage of reliable epidemiological data, little public awareness, and a critical need for accurate diagnosis and clinical care of HBV infection, most importantly among asymptomatic and apparently healthy individuals in Nigeria, particularly regarding IEMs and DRMs.

The majority of asymptomatic people are unaware carriers; this tends to increase the continuous spread and transmission within the community. Possible modes of transmission include but are not limited to blood transfusions, vertical transmission from mother to child during childbirth, injecting drug use, re-use of contaminated needles and syringes, and sexual contact [25,26]. The risk of asymptomatic infection with HBV is enormous and has long-term implications for disease elimination and eradication. Most importantly, infection with HBV undetected by the regular and most common testing assay in blood donors may be transmitted by blood transfusion. Therefore, active genomic surveillance of HBV infection is important to effectively control the disease.

This study provides insight into the possible emergence and spread of IEMs and BCP/PC mutants in asymptomatic HBV genotype E carriers in southwestern Nigeria.

## 2. Materials and Methods

### 2.1. Ethical Consideration and Clinical Sample Collection

This research was approved by the State Specialist Hospital Health Research and Ethics Committee, Osun State (HREC/27/04/2019/SSHO/502), Ekiti State University Teaching Hospital Ethical Research Committee (EKSUTH/A67/2019/08/003), and the Ladoke Akintola University of Technology (LAUTECH) Hospital Research Ethics Committee (LTH/2019/12/438). Before sample collection, all enrolled participants provided their written informed consent. Blood samples were obtained from 410 individuals in the southwestern Nigerian states of Osun and Ekiti between 2019 and 2021. At the time of blood collection, all the study participants were from a cohort of either blood donors, outpatients, or antenatal patients at the hospital’s medical outpatients’ units of the selected hospitals in Osun and Ekiti states and apparently asymptomatic for liver diseases. The serological profile is based on ELISA testing of this study’s cohorts’ HBsAg elsewhere [27]. All samples were tested for six HBV serological markers, including hepatitis B virus surface antigen (HBsAg), hepatitis B virus surface antibody (HBsAb), hepatitis B virus e antigen (HBeAg), antibody to hepatitis B virus E antigen (anti-HBe), hepatitis B core antigen to immunoglobulin M (HBcAb-IgM), and hepatitis B core total (antiHBc) using commercially available enzyme-linked immunosorbent assay (ELISA) kits (Melsin Medical Co., Changchun, China) following the manufacturer’s instruction.

### 2.2. HBV Genomic DNA Extraction

Total HBV-DNA were extracted from 200ul of serum from the four hundred and ten (410) patients’ sera using a Qiagen DNEasy kit (Lot number 163049045; Qiagen, Germany), according to the manufacturer’s instructions. DNA was eluted in 50 µL and stored at −80 °C until further use.

### 2.3. PCR Amplification of the HBV S-Gene

The nested PCR technique was used for the amplification of the fragments of the HBV S gene according to the method of Forbi and others (2010) and Umego and others (2022). The 400 bp fragment of the partial “S” gene region was amplified using the PuReTaqTM Ready-To-Go PCR Beads in strip tubes (Sigma-Aldrich^®^ Darmstadt, Germany Lot # 17262518). The position of the codons in the S gene covered by the PCR ranges from 22 to 170. The PCR primers used were HBV_S1F (5”-CTAGGACCCCTGCTCGTGTT-3’) and HBV_S1R (5′-CGAACCACTGAACAAATGGCACT-3’) for the first round, and the second-round primers were HBV_SNF (5′-GTTGACAAGAATCCTCACAATACC-3’) and HBV_SNR (5′-GAGGCCCACTCCCATA-3). First- and second-round PCR reaction conditions were similar, except that extracted DNA was used as the template for the first-round PCR, while the first-round PCR products were used as a template for the second-round PCR amplification. The PCR amplification was carried out in a 20 µL reaction by adding 13 µL RNase-free water, 1µL of each of the primers (made in 20 µM concentrations), and 5 µL of a DNA template into the strip tube containing the puReTaq beads. A PCR procedure was carried out using an Eppendorf Thermal cycler (Eppendorf, Stevenage, UK) as follows: 94 °C for 3 min, followed by 45 cycles of denaturation, 94 °C for 30 s, annealing, 55 °C for 60 s and elongation, and 70 °C for 40 s with a ramp of 40% from 55 °C to 70 °C. The reaction was further elongated at 72 °C for 7 min and held at 4 °C until the reaction was terminated. PCR products were resolved on 2% agarose gel stained with ethidium bromide and viewed using a UV transilluminator.

### 2.4. PCR Amplification BCP/PC Genome Region

The BCP/PC region was amplified based on a previously reported nested PCR protocol targeting a 360bp fragment of the BCP/PC region, as described by [28]. Briefly, DNA amplification was performed using puReTaqTM Ready-To-Go PCR Beads in strips (Sigma-Aldrich^®^ Lot # 17262518). The first round of PCR amplification was carried out using a 20 µL reaction, adding 16µL of RNase-free water, 1 µL of each of the primers (made in 20 µM concentrations), and 2 µL of a DNA template into the strip tube containing the puReTaq bead. The first round of PCR primers used are BCP_PC F1 (5-GCATGGAGACCACCGTGAAC-3) and BCP_PC R1 (5GGAAAGAAGTCCGAGGGCAA-3). The thermal cycling conditions were an initial denaturation of 30 s at 94 °C, followed by 55 °C at 60 s, 35 cycles of 60 s at 72 °C denaturation, 2 min at 72 °C annealing, 30 s at 68 °C extension, and a final extension for 10 min at 68 °C. The second round of PCR amplification was carried out using a 20 µL reaction, adding 16 µL of RNase-free water, 1 µL of each of the primers (made in 20 µM concentrations), and 2µL of DNA template from the first round of PCR. The second round of PCR primers used are BCP_PC F2 (5- CATAAGAGGACTCTTGGACT-3 and BCP_PC R2 (5- GGCAAAAAACAGAGTAACTC-3). The cycling conditions were an initial denaturing of 94 °C for 3 min, followed by 30 cycles (denaturation, 94 °C for 30 s, annealing, 55 °C for 30 s and elongation, 72 °C for 60 s with a ramp of 40% from 55 °C to 70 °C). The reaction was further elongated at 72 °C for 2 min and held at 4 °C until the reaction was terminated. The PCR procedure used an Eppendorf Thermal cycler (Eppendorf, UK). The PCR products were resolved in 2% agarose gel stained with ethidium bromide and viewed using a UV transilluminator.

### 2.5. Sequencing of the S and BCP/PC Genes Using the Sanger Sequencing Method

Five microliters of the secondary amplicons from the HBS gene and BCP/PC gene with the expected band size (400 bp and 360 bp, respectively, codon positions 22–170 and 54–271, respectively) were purified with 2 uL of ExoSAP-IT (Affymetrix, Santa Clara, CA, USA) at 37 °C for 15 min and inactivated at 80 °C for 15 min. For sequencing prep, 4 uL of big dye terminator 3.1 ready reaction mix (Life Technologies, Carlsbad, CA, USA), 7 uL of nuclease-free water, 2 uL each of secondary PCR (HBV_SNF and HBV_SNR), and (BCP_PCF1 and BCP_PCF2) primers were added to the purified secondary amplicons. PCR amplification was performed on a thermocycler (Eppendorf Vapo. Protect Mastercycler pro, Germany) in a final volume of 20 μL at cycling conditions of 96 °C for 60 s, 30 cycles 96 °C for 10 s, 50 °C for 5 s, and 70 °C for 4 min. The sequencing products were further purified using the bigdye Xterminator kit (Life Technologies, Carlsbad, CA, USA) (90 uL of SAM solution and bigdye Xterminator bead solution per sample) vortexed for 20 min at room temperature. Sanger sequencing was performed on the Applied Biosystems 3500 XL series Genetic Analyzer at the African Centre of Excellence for Genomics of Infectious Diseases, Redeemer’s University, Nigeria.

### 2.6. Phylogenetic and Gene Variability Analysis of HBV Partial S Gene and BCP/PC Gene Sequences

The forward and reverse sequences per sample were stitched into contigs for the HBV S and BCP/PC genes. Each contig was subjected to a Blastn search on the National Centre for Biotechnology Information (NCBI) page. Aligning nucleotide sequences and deduced amino acids with those of reference HBV genotypes A-H from the NCBI virus was performed using the MAFFT online service [29]. Phylogenetic trees were constructed using IQ-TREE version 1.6.12 [30] with ModelFinder [31] and ultrafast bootstrap (1000 replicates) [32]. The tree was visualized using Interactive Tree of Life (iTOL) v5 [33].

For the S gene ORF, we examined both the major hydrophilic region (MHR) (amino acid position 99-169) and the a-determinant domain (found within the MHR; amino acid position 124-147) using BioEdit and geno2pheno software (https://hbv.geno2pheno.org/ accessed on 25 March 2023). All our sequences were aligned with the HBV genotype E reference sequence (accession number LC513651) from the NCBI.

During BCP/PC sequences analysis, we examined nucleotide changes at the TA-rich genome regions, including nt 1750-1755, 1758-1762, 1771-1775, and 1778-1795. Kozak sequence mutants (translational genes-nt1809-1812), PC initiation (1814-1816), the post-translational mutant gene (G1862T), and the translational stop codon (G1896A with C1858T) were used [28,34]. The partial basal core promoter (BCP) and precore (PC) region gene sequences obtained from this study were aligned with the previously reported wild type (X75657.1) and mutant (AF28996.1).

### 2.7. Statistical Analysis

A chi-squared test was performed on categorical data to test for significant correlations between the different factors using SPSS version 21 (IBM Corp. Released 2012. IBM SPSS Statistics for Windows, Version 21.0. Armonk, NY, USA). A less than 0.05 *p*-value was used to determine statistical significance.

## 3. Results

### 3.1. Socio-Demographic Characteristics of the Respondents

In this study, 150 samples were from Ekiti and 260 were from Osun. The mean age of the 410 participants included in the study was 36.77 ± 11.26, and 36.8% of the respondents were between 25 and 34 years of age. A total of 283 (69%) respondents were female, while 127 (21%) were male. A total of 308 (75.1%) of the 410 participants were married. A total of 100 (24.4%) are single, and 2 (0.5%) are divorced. Those who had completed graduate studies were 184 (44.9%), while 221 (53.9%) of the respondents were self-employed (Table 1).

### 3.2. HB S and BCP/PC Gene-Specific PCR Amplification Results

The 410 samples were initially screened by ELISA, of which 51 (12.4%) were positive to HBsAg and 12 (6.6%) were positive to HBeAg. The results of the serological profile are published elsewhere [27]. HBV (HB gene- and/or BCP/CP gene-positive PCR) was detected in 12.4% (*n* = 51/410) of individuals who were also HbsAg-positive by ELISA. The mean age of the 51 samples is 36.08 ±8.03. Of the 51 HBV-PCR-positive samples, 60.8% (*n* = 31) were from Osun, while 39.2% (*n* = 20) were from Ekiti. Of the 51 HBV DNA-positive samples, 32 (62.7%) and 19 (37.3%) were detected in females and males, respectively. According to the categories of individuals, 43.1% (*n* = 22) were from blood donors, 41.2% (*n* = 21) were from outpatients, and 15.7% (*n* = 8) were from pregnant women. Seven (13.7%) were positive to HbeAg, while forty-four (86.3%) were negative to HBeAg, respectively (Table 2, Appendix A).

### 3.3. Phylogenetic Analysis of the HBV S and BCP/PC Genes

PCR amplification of the S gene was successful in 24 (47%) of the 51 isolates that tested positive for HBsAg. Results from both geno2pheno and phylogenetic analysis using IQTREE showed that all the 24 HBV S gene sequences obtained in this study were classified as HBV genotype E and clustered around samples from Nigeria, Guinea, and the Central African Republic (Figure 1).

### 3.4. Analysis of Immune Escape Mutations Present in the HBV Samples

Of the 410 samples screened by PCR, 51 (12.4%) were positive for HB gene amplification and were sequenced. Only 47.1% (*n* = 24) of the S gene amplicon sequences were of good quality for downstream analysis. Mutational analysis of the MHR and a-determinant domain of S gene sequences in this study revealed the presence of three immune escape mutations in 37.5% (9/24) of the samples: two samples (8.3%) (NGR-HBV-OS239 and NGR-HBV-OS68) had a T116N substitution, six samples (25%) (NGR-HBV-OS239, NGR-HBV-OS68, NGR-HBV-OS155, NGR-HBV-OS242, NGR-HBV-OS216, and NGR-HBV-OS245) had heterogenous D144A/N/S/H substitution, and one sample (4.2%) (NGR-HBV-OS155) had a G145E substitution (Figure 2).

### 3.5. Analysis of the Basal Core Protein/Precore (BCP/PC) Genome Variability in the HBV Samples

Of the 51 BCP/PC PCR-positive samples sequenced, 39 (76.5%) sequences were of good quality for mutational analysis. The mutational analysis of the BCP/PC region revealed various substitutions in the TA-rich genome regions. Specifically, we observed a T1753G/C in two samples (NGR-HBV-OS245 and NGR-HBV-EK37), while samples NGR-HBV-OS2 and NGR-HBV-EK73 both had T1754A and A1755G substitutions, respectively (Figure 3A). We equally observed a preponderance of major BCP mutants with the prevalence of BCP double substitutions ranging from 38.5% (A1762T) to 43.6% (G1764A). Interestingly, the BCP double mutants were highest among the outpatient group, as well as among individuals with HBeAg-negative status (33.3% and 38.5%) (Table 3). Analysis of the presence of Kozak sequence mutants (nt 1809-1812) showed that two samples (NGR-HBV-EK105 and NGR-HBV-OS80) had a G1809C/A substitution, while one sample (NGR-HBV-OS49) had an A1811C substitution (Figure 3B). Analysis of the PC initiation region (nt1814-1816) revealed one sample (NGR-HBV-EK105) having an A1814C substitution. No post-translational mutant gene (G1862T) was detected in our study (Figure 4). Previously reported classical PC mutant variants were observed in high proportion, including G1896A (33.3%) and G1899A (12.8%), and were equally higher among individuals with HbeAg-negative status (20.5% and 7.7%) (Figure 4).

The HBV sequences generated in this study were deposited into the NCBI GenBank under accession numbers OR001858, OR001861, OR001862, OR001859, OR001869, OR001876, OR001865, OR001877, OR001878, OR001879, OR001870, OR001871, OR001866, OR001880, OR001872, OR001873, OR001874, OR001875, OR001860, OR001881, OR001867, OR001863, OR001864, and OR001868.

## 4. Discussion

The prevalence of HBV infection in Nigeria is between 10% and 15%. Thus, it is a major public health concern, despite the availability of the HBV vaccine for adults and children [35]. Vaccine- and diagnostic-escape phenomena have reportedly been caused by immune escape mutations in general [36]. This study documents the prevalence of immune escape mutants (IEMs) and BCP/PC HBV mutants circulating among southwestern Nigeria’s HBV genotype E-infected cohort without symptoms. All the HBV isolates analyzed in this study were all genotype E of the virus. This is not surprising, as this is the most common virus genotype reported in the country [37]. Other genotypes reported in Nigeria are A, B, C, and D [36,38,39]. Specifically, we identified important substitutions in the “a”-determinant domain and clinically relevant mutations in the BCP and PC HBV genomic regions in asymptomatic hospital attendees. The results of this study showed the presence of three IEMs: T116N, D144A/N/S/H, and G145E, which have been linked to decreased antibody binding to wild type S protein by virions and subviral particles, resulting in breakthrough infections and diagnostic failure [40,41,42,43].

The prevalence of HBV IEMs in the restricted, immunodominant “a”-determinant region (ADR) between amino acid positions 124–147 was 37.5% (*n* = 9/24), where nine connotes the number of samples with IEM mutations occurring in the antigenic determinant region. The prevalence of HBV IEMs in this study was slightly higher than what was earlier reported by Osasana in Nigeria and Lazarevec in Brazil, where a prevalence of 29% and 10.7% were reported, respectively [39,44]. This high prevalence may be due to the fact that the study participants were asymptomatic and, therefore, have yet to undergo HBV infection treatment. This study further corroborates the circulation of IEMs in Nigeria. However, the pattern of IEMs documented in this study differs from previously documented dominant Q129H [36] and G145K [23,45] IEMs in Nigeria. Other notable non-immune escape mutations detected in the MHR domain were L104W (1), T115N (3), and S117N/K (3).

Interestingly, all the IEMs were detected in HBV sequences from outpatients and apparently healthy prospective blood donors in Osun state. Previously detected IEMs in Nigeria were documented in different populations, including apparently healthy community dwellers, pregnant women, and prospective blood donors in Ekiti, Ondo, and Oyo [36,44]. The spread of these IEMs in Nigeria may jeopardize HBV vaccine efficacy and have public health implications in managing and controlling the infection if not addressed. The varying patterns of IEMs imply the continuous evolution of IEMs in the region; thus, there is a need for more robust HBV surveillance in different parts of the country to ascertain their true prevalence and identify factors that may enhance their transmission.

Mutations within and outside the “a”-determinant domain have been linked to Occult HBV Infection (OBI) [46,47,48]. Substitutions that result in hydrophobicity, the presence of a phenyl group, and charges (previously absent) in the side chain of amino acid residues located in the MHR may enhance immune escape strain adaptation and circulation via decreased HBsAg secretion and impaired reactivity with anti-HBs antibodies among the asymptomatic study cohort [47].

A high proportion of classical BCP/PC mutant variants identified in the current study had been previously associated with HBeAg-negative CHB infection and the down-regulation of PC mRNA transcription [49]. The classical BCP double mutations A1762T/G1764A (38.5%/43.6%) were the most common and were equally highest among the outpatient group (23.1%/28.2%) and HBeAg-negative participants (33.3%/38.5%), respectively (Table 3). The classical BCP double mutations A1762T/G1764A were the most common (Table 3). Additionally, the G to A point mutation at nucleotide 1896 of the precore region results in the conversion of tryptophan (TGG) to a stop codon (TAG), which is known to abolish HBeAg synthesis [13,50,51], was found to be 33.3% in the HBV genotype E-infected cohort in this study. The frequency of BCP/PC mutations found in this study is similar to the previously reported prevalence found in the ART-naïve HIV/HBV co-infected genotype E-infected cohort in Nigerians and calls for robust HBV surveillance in the country [52]. Regardless of the study group, the majority of classical BCP/PC mutants, such as BCP double mutant sequences, PC initiation, and G1896A with G1899A mutations, were significantly more common in HbeAg-negative subjects. The limitation of this study is that we could not explain the observed high proportion and implications of classical BCP/PC mutant variants identified across the study subjects because participants’ liver disease status and HBV viral load were not assessed due to financial constraints.

However, the continuous spread of the IEMs and BCP/PC mutations among asymptomatic patients may contribute to vaccine and diagnostic failure [53,54] and the evolution of new strains. Molecular surveillance of circulating HBV must be conducted within the country to guide the implementation of effective management and control methods.

Some implications of these mutations observed among asymptomatic patients are the ability to easily transmit the virus within the community, as most patients are unaware carriers of HBsAg, and the mutations. These findings from our study also emphasize the risk of hepatitis B virus transmission through blood donation or transfusion. These findings also been implicated in cryptogenic liver diseases, progressive acute and chronic HBV infection, and hepatocellular carcinoma development. It is, therefore, imperative to conduct more in-depth epidemiological studies on the success or failure of HBV vaccination programs in Nigeria and Africa and ensure all vaccinated individuals complete the expected dose of vaccine.

In conclusion, a high frequency of IEM, BCP, and PC HBV mutants were found in asymptomatic outpatients from southwestern Nigeria’s HBV genotype E-infected cohort. This suggests the increased likelihood of potential HBeAg-negative chronic HBV infection and the emergence of HBV strains, which may result in breakthrough infections and diagnostic failure in this setting. The IEM pattern observed in this study differed from previously documented dominant IEMs in Nigeria, implying the continuous evolution of IEM strains in the country. More research is needed to determine their prevalence and identify factors contributing to their spread. There is a need for active serological and molecular surveillance of the circulating HBV and mutations on its transmission, spread, and effect on asymptomatic carriers, as well as its interactions with the existing vaccine.

## Figures and Tables

**Figure 1 viruses-15-02188-f001:**
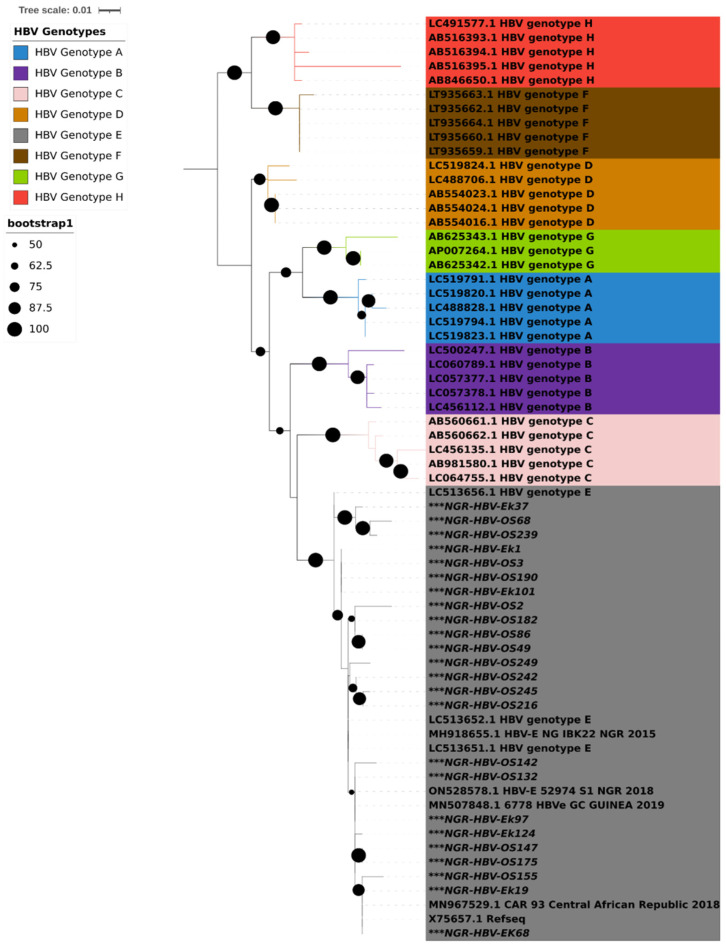
Maximum likelihood tree with ModelFinder based on partial S gene sequences with 1000 bootstrap replications. All HBV genotypes are color coded, as shown in the legend. HBV sequences reported in this study are asterisked (***) in black. The Interactive Tree of Life (iTOL) v5 with midpoint rooting was used to visualize the tree.

**Figure 2 viruses-15-02188-f002:**
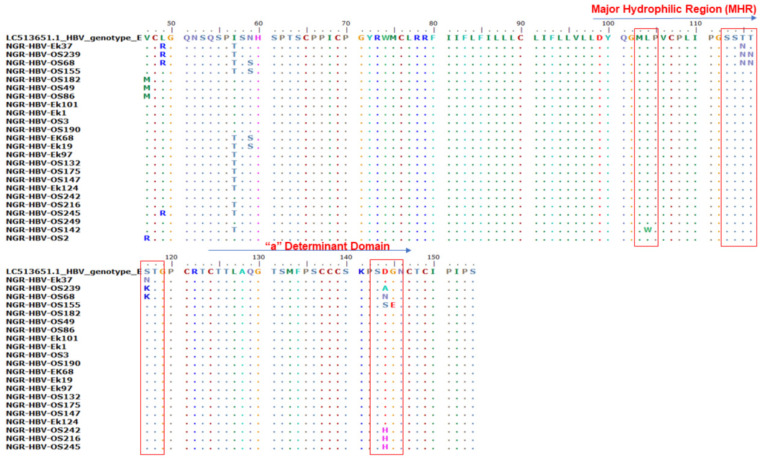
Alignment of the partial S gene sequences with the HBV genotype E reference sequence (accession number LC513651). The various mutations at the major hydrophilic region (amino acid position 99–169) and the a-determinant domain found within the MHR (amino acid position 124–147).

**Figure 3 viruses-15-02188-f003:**
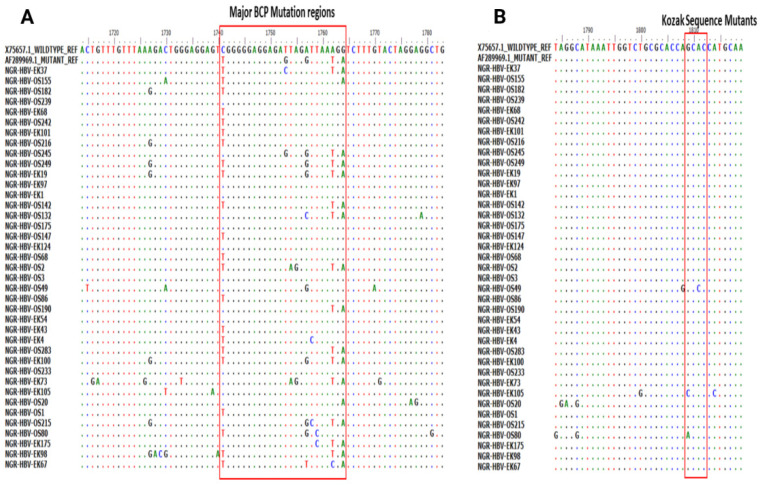
(**A**) Partial Basal core promoter (BCP) region gene sequences obtained from this study and aligned with previously reported wildtype (X75657.1) and mutant (AF28996.1) Alignment showed the various mutations at the BCP region with empahsis toon the G1896/G1899A vaiant and (**B**) various mutations at the Kozak sequence region (nt 1809-1812).

**Figure 4 viruses-15-02188-f004:**
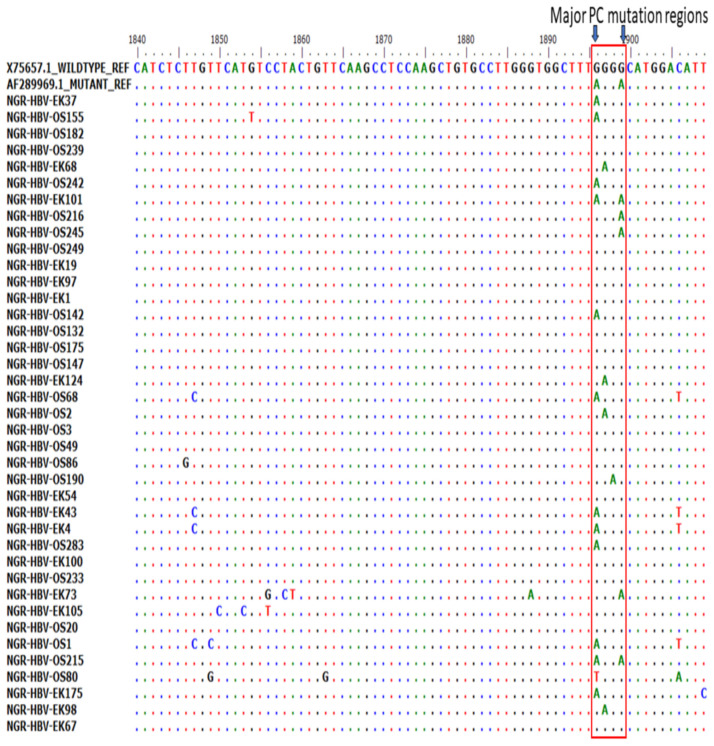
Precore (PC) region gene sequences obtained from this study aligned with previously reported wild type (X75657.1) and mutant (AF28996.1). Results showed the presence of various mutations at the PC region with emphasis on G1896/G1899A variants.

**Table 1 viruses-15-02188-t001:** Socio-demographic characteristics of the total respondents.

N	51
**Age**	
Mean	36.08 ± 8.03
Range	22
**Gender**	
Female	19 (37.3%)
Male	
**Location**	
Ekiti	20 (39.2%)
Osun	31(60.8%)
**Study Cohorts**	
Blood Donor	22 (43.1%)
Outpatient	21 (41.2%)
Perinatal	8 (15.7%)
**HBeAg Status**	
Negative	44 (86.3)
Positive	7 (13.7)

**Table 2 viruses-15-02188-t002:** Demographic profile of participants with HBV-positive PCR outcomes analyzed in this study.

N	51
**Age**	
Mean	36.08 ± 8.03
Range	22
**Gender**	
Female	19 (37.3%)
Male	
**Location**	
Ekiti	20 (39.2%)
Osun	31(60.8%)
**Study Cohorts**	
Blood Donor	22 (43.1%)
Outpatient	21 (41.2%)
Perinatal	8 (15.7%)
**HBeAg Status**	
Negative	44 (86.3)
Positive	7 (13.7)

**Table 3 viruses-15-02188-t003:** Frequency distribution of BCP and PC mutation among HBeAg status and study population.

	Mutation	HBeAg Status *n* (%)			*p*-Value	Study Population *n* (%)				*p*-Value
BCP Mutations		Positive (*n* = 6)	Negative (*n* = 33)	Total (*n* = 39)		Perinatal (*n* = 7)	Blood Donor (*n* = 14)	HIV Co-Infected (*n* =18)	Total (*n* = 39)	
	C1675T	0 (0.0)	1 (2.6)	1 (2.6)	0.66	0 (0.0)	1 (2.6)	0 (0.0)	1 (2.6)	0.40
	C1678T/A	0 (0.0)	1 (2.6)	1 (2.6)	0.66	0 (0.0)	1 (2.6)	0 (0.0)	1 (2.6)	0.40
	C1703A	0 (0.0)	2 (5.1)	2 (5.1)	0.53	0 (0.0)	1 (2.6)	1 (2.6)	2 (5.1)	0.77
	A1727G	1 (2.6)	7 (17.9)	8 (20.5)	0.80	2 (5.1)	3 (7.7)	3 (7.7)	8 (20.5)	0.79
	C1730A/T/G	0 (0.0)	4 (10.3)	4 (10.3)	0.36	1 (2.6)	0 (0.0)	3 (7.7)	4 (10.3)	0.28
	C1741T	4 (10.3)	18 (46.2)	22 (56.4)	0.58	5 (12.8)	9 (23.1)	8 (20.5)	22 (56.4)	0.36
	T1753C/G	0 (0.0)	2 (5.1)	2 (5.1)	0.53	0 (0.0)	2 (5.1)	0 (0.0)	2 (5.1)	0.15
	A1757G/C/T	2 (5.1)	7 (17.9)	9 (23.1)	0.51	0 (0.0)	4 (10.3)	5 (12.8)	9 (23.1)	0.27
	T1758C	1 (2.6)	1 (2.6)	2 (5.1)	0.16	0 (0.0)	1 (2.6)	1 (2.6)	2 (5.1)	0.77
	A1762T	2 (5.1)	13 (33.3)	15 (38.5)	0.77	1 (2.6)	5 (12.8)	9 (23.1)	15 (38.5)	0.24
	G1764A	2 (5.1)	15 (38.5)	17 (43.6)	0.58	1 (2.6)	5 (12.8)	11 (28.2)	17 (43.6)	0.08
	T1771C	0 (0.0)	1 (3.0)	1 (2.6)	0.66	0 (0.0)	0 (0.0)	1 (2.6)	1 (2.6)	0.55
	T1809G/C/A	1 (2.6)	1 (2.6)	2 (5.1)	0.16	0 (0.0)	0 (0.0)	2 (5.1)	2 (5.1)	0.29
	A1811C	0 (0.0)	1 (2.6)	1 (2.6)	0.66	0 (0.0)	0 (0.0)	1 (2.6)	1 (2.6)	0.55
	A1814C	0 (0.0)	1 (2.6)	1 (2.6)	0.66	0 (0.0)	0 (0.0)	1 (2.6)	1 (2.6)	0.55
**PC Mutations**	T1847C	1 (2.6)	3 (7.7)	4 (10.3)	0.57	1 (2.6)	3 (7.7)	0 (0.0)	4 (10.3)	0.13
	T1850C	0 (0.0)	1 (2.6)	1 (2.6)	0.66	0 (0.0)	0 (0.0)	1 (2.6)	1 (2.6)	0.55
	T1858C	0 (0.0)	1 (2.6)	1 (2.6)	0.66	0 (0.0)	0 (0.0)	1 (2.6)	1 (2.6)	0.55
	G1888A/T	0 (0.0)	1 (2.6)	1 (2.6)	0.66	0 (0.0)	0 (0.0)	1 (2.6)	1 (2.6)	0.55
	G1896A	5 (12.8)	8 (20.5)	13 (33.3)	0.005 *	1 (2.6)	5 (12.8)	7 (17.9)	13 (33.3)	0.49
	G1899A	2 (5.1)	3 (7.7)	5 (12.8)	0.102	0 (0.0)	2 (5.1)	3 (7.7)	5 (12.8)	0.52

* *p*-value significant at <0.05.

## Data Availability

Genome sequences of HBV reported in this study have been deposited in GenBank under accession numbers OR001858–OR001881.

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
