# Peer review of "Detection of Immune Escape and Basal Core Promoter/Precore Gene Mutations in Hepatitis B Virus Isolated from Asymptomatic Hospital Attendees in Two Southwestern States in Nigeria"

_viruses, 2023, doi:10.3390/v15112188_

Round 1

Reviewer 1 Report

Comments and Suggestions for Authors

Sobajo et al. reported the prevalence of HBV mutants (S gene MHR, BCP-PC region) among 410 HBV carriers in southwest Nigeria. This study was an exploratory investigation that demonstrated the molecular characteristics of HBV in the West African population. Their data provides valuable reference materials for the HBV research field. However, the paper did not present many new findings, and several issues should be addressed, particularly for the data presentation and analysis.

Major comments

(1) The BCP/PC mutation ratio is highly associated with the HBeAg status; the author should re-group the cohort according to the serological HBeAg tests. In the original manuscript, I do not find the HBeAg data.

(2) The authors screened a total of samples from 410 people. Are all these samples HBsAg positive? The authors did not show any serological data in this manuscript, as they declared that the serological profiles of the cohort had been published elsewhere. However, because of the potential relationship between serological profiles and viral mutation characteristics (e.g., anti-HBs [MHR mutations], HBeAg [BCP/PC mutations], HBsAg level), it cannot accurately describe the molecular characteristics without considering the serological profiles.

(3) There were no scientifically valuable conclusions from this manuscript's data and results.

Minor comments

(1)   Table 1 is not formatted properly. “Educational status” should be centered. The number in the “Mean±SD” row should be left-justified.

(2)   Figure 2, the “MHR “a” determinant domain” is a part of the MHR

(3)   The circles around labels A and B in Figure 3 should be removed.

Author Response

Sobajo et al. reported the prevalence of HBV mutants (S gene MHR, BCP-PC region) among 410 HBV carriers in southwest Nigeria. This study was an exploratory investigation that demonstrated the molecular characteristics of HBV in the West African population. Their data provides valuable reference materials for the HBV research field. However, the paper did not present many new findings, and several issues should be addressed, particularly for the data presentation and analysis.

 Major comments

(1) The BCP/PC mutation ratio is highly associated with the HBeAg status; the author should re-group the cohort according to the serological HBeAg tests. In the original manuscript, I do not find the HBeAg data.

Response: Thank you very much for this comment. We have now added the serology result for the HBeAg test in the manuscript in line 223 to 224 ‘The 410 samples were initially screened by ELISA, of which 51(12.4%) was positive to HBsAg while 12(6.6%) was positive to HBeAg’ and also on the supplementary table 2. Of the 51 positive samples to HBsAg,  7(13.7%) were positive to HBeAg while 44(86.3%) were negative to HBeAg respectively in line 231- 232. (Table 2).

(2) The authors screened a total of samples from 410 people. Are all these samples HBsAg positive? The authors did not show any serological data in this manuscript, as they declared that the serological profiles of the cohort had been published elsewhere. However, because of the potential relationship between serological profiles and viral mutation characteristics (e.g., anti-HBs [MHR mutations], HBeAg [BCP/PC mutations], HBsAg level), it cannot accurately describe the molecular characteristics without considering the serological profiles.

Response: thank you very much for this comment. Of the 410 samples, only 51 samples were positive to HBsAg by ELISA and also by PCR. We have now added Supplementary table 1 which consists of the  Prevalence of HBV serological biomarkers by socio-demographic characteristics of participants which has been previously published elsewhere.

(3) There were no scientifically valuable conclusions from this manuscript's data and results.

Response: we have included BCP Mutation in relation to HBeAg in line 274 to 275 ‘ Interestingly, the BCP double mutants were highest among the outpatient’s group as well as among individuals with HBeAg negative status (33.3% and 38.5%) (Table 3) and  Previously reported classical PC mutant variants were observed in high proportion, including G1896A (33.3%) and G1899A (12.8%) and were equally higher among individuals with HBeAg negative status (20.5% and 7.7%) (Figure 4).’in line 281 to 282. We have also included the HBeAg status in table 3. 

In the Discussion section, we have now added “ A high proportion of classical BCP/PC mutant variants identified in the current study had been previously associated with HBeAg-negative CHB infection and the down-regulation of PC mRNA transcription [49]. The classical BCP double mutations A1762T/G1764A (38.5%/43.6%) were the most common and were equally highest among out-patients group (23.1%/28.2%) and HBeAg-negative participants (33.3%/38.5%) respectively (Table 3).” in line 342-346

We also addedAdditionally, the G to A point mutation at nucleotide 1896 of the precore region results in the conversion of tryptophan (TGG) to a stop codon (TAG) which is known to abolish HBeAg synthesis” in line 347-349.

We have also includedRegardless of the study group, the majority of classical BCP/PC mutants, such as BCP double mutant sequences, PC initiation, and G1896A with G1899A mutations, were significantly more common in HBeAg negative subjects” in line 353-355

 We have added “This suggests the increased likelihood of potential HBeAg-negative chronic HBV infection and the emergence of HBV strains that may result in breakthrough infections and diagnostic failure in this setting.” to the conclusion in line 373-376.

Minor comments

(1)   Table 1 is not formatted properly. “Educational status” should be centered. The number in the “Mean±SD” row should be left-justified.

Response: Thank you very much for this comment, we have now formatted Table 1 properly.

(2)   Figure 2, the “MHR “a” determinant domain” is a part of the MHR

Response: Thank you very much for this comment,  we have now removed the “MHR” from the “a determinant domain” since it’s a part of the MHR.

 (3)   The circles around labels A and B in Figure 3 should be removed.

Response: Thank you very much for your response, we have now removed the circles around label A and B

Reviewer 2 Report

Comments and Suggestions for Authors

This is a good and well written paper. Just something for the authors:

It would be interesting to know, if these data are available, the viral loads detected in the blood samples used in the experiments. If the data are not available the reasons why shoulbe stated in the text

The same authors obtained simular results in nigerian individuals also with different genotype: this does not reduce the value of this paper, but these researches should be mentioned in the text

Patterns of hepatitis b virus immune escape and pol/rt mutations across clinical cohorts of patients with genotypes a, e and occult hepatitis b infection in Nigeria: A multi-centre study Oluwadamilola G.

Detection of hepatitis B virus isolates with mutations associated with immune escape mutants among pregnant women in Ibadan, southwestern Nigeria Faleye T.O.,  2015 

Since the mentioned mutations may affect HBV conformation and this may be relevant from a pathogenetic point of view, we think that this aspect deserves some comment and reference

Khodadad N. In silico functional and structural characterization of hepatitis B virus PreS/S-gene in Iranian patients infected with chronic hepatitis B virus genotype D. Heliyon. 2020 

Di Stefano M, Faleo G, Leitner T, Zheng W, Zhang Y, Hassan A, Alwazzeh MJ, Fiore JR, Ismail M, Santantonio TA. Molecular and Genetic Characterization of Hepatitis B Virus (HBV) among Saudi Chronically HBV-Infected Individuals. Viruses. 2023 Feb 6;15(2):458.

Author Response

Comments and Suggestions for Authors

This is a good and well written paper. Just something for the authors:

  1. It would be interesting to know, if these data are available, the viral loads detected in the blood samples used in the experiments. If the data are not available the reasons why should be stated in the text

Response: Thank you very much, the HBV viral load was not detected due to financial constraint. This was one of the limitation in line 358 to 359 ‘The limitation of this study is that we could not explain the observed high proportion and implications of classical BCP/PC mutant variants identified across the study subjects in the current study because participants' liver disease status and HBV viral load were not assessed due to financial constraints’

  1. The same authors obtained simular results in nigerian individuals also with different genotype: this does not reduce the value of this paper, but these researches should be mentioned in the text

 Patterns of hepatitis b virus immune escape and pol/rt mutations across clinical cohorts of patients with genotypes a, e and occult hepatitis b infection in Nigeria: A multi-centre study Oluwadamilola G.

Response: Thank you very much for this response, we have now captured this study from Osasona et al., 2023 in line 311, 320 to 321

Detection of hepatitis B virus isolates with mutations associated with immune escape mutants among pregnant women in Ibadan, southwestern Nigeria Faleye T.O.,  2015 

Response: Thank you very much for this response, Faleye et al., 2015 was referenced in line 331

  1. Since the mentioned mutations may affect HBV conformation and this may be relevant from a pathogenetic point of view, we think that this aspect deserves some comment and reference

 Khodadad N. In silico functional and structural characterization of hepatitis B virus PreS/S-gene in Iranian patients infected with chronic hepatitis B virus genotype D. Heliyon. 2020 

Di Stefano M, Faleo G, Leitner T, Zheng W, Zhang Y, Hassan A, Alwazzeh MJ, Fiore JR, Ismail M, Santantonio TA. Molecular and Genetic Characterization of Hepatitis B Virus (HBV) among Saudi Chronically HBV-Infected Individuals. Viruses. 2023 Feb 6;15(2):458.

Response: Thank you very much for this response, we have now referenced khadada N  and Di Stefano in line 361

Reviewer 3 Report

Comments and Suggestions for Authors

This study provided the prevalence of immune escape mutants and BCP/PC mutants among South-western Nigeria's HBV genotype E infected cohort but without symptoms. This is an interesting study. However, there are some issues.

1. All these patients were HBsAg-positive. Please explain the method of HBsAg assay in this article, including the limit of detection for HBsAg.

2. Could you provide HBsAg levels and HBV DNA levels for these patients?

3. The samples were initially screened by ELISA and the results of the serological profile were published in Reference 27. Is there HBV/HDV co-infection among the 51 individuals with detectable HBV DNA?

Comments on the Quality of English Language

This study provided the prevalence of immune escape mutants and BCP/PC mutants among South-western Nigeria's HBV genotype E infected cohort but without symptoms. This is an interesting study. However, there are some issues.

1. All these patients were HBsAg-positive. Please explain the method of HBsAg assay in this article, including the limit of detection for HBsAg.

2. Could you provide HBsAg levels and HBV DNA levels for these patients?

3. The samples were initially screened by ELISA and the results of the serological profile were published in Reference 27. Is there HBV/HDV co-infection among the 51 individuals with detectable HBV DNA?

Author Response

Comments and Suggestions for Authors

This study provided the prevalence of immune escape mutants and BCP/PC mutants among South-western Nigeria's HBV genotype E infected cohort but without symptoms. This is an interesting study. However, there are some issues.

  1. All these patients were HBsAg-positive. Please explain the method of HBsAg assay in this article, including the limit of detection for HBsAg.

Response: Thank you very much for this response. 51 samples out of the 410 were positive to HBsAg by ELISA and conventional PCR using gel Electrophoresis. The method for the ELISA assay has been added in line 115 to 120 ‘All samples were tested for six HBV serological markers, including Hepatitis B virus surface antigen (HBsAg), Hepatitis B virus surface antibody (HBsAb), Hepatitis B virus e antigen (HBeAg), antibody to Hepatitis B virus e antigen (anti-HBe), Hepatitis B core antigen to immunoglobulin M (HBcAb-IgM) and Hepatitis B core total  (antiHBc) using commercially available enzyme-linked immunosorbent assay (ELISA) kits (Melsin Medical Co., China). 

  1. Could you provide HBsAg levels and HBV DNA levels for these patients?

Response: Thank you very much for the response, The DNA was not quantified as the PCR was not a quantitative PCR but the HBV DNA was determined by Conventional PCR using be gel electrophoresis.

  1. The samples were initially screened by ELISA and the results of the serological profile were published in Reference 27. Is there HBV/HDV co-infection among the 51 individuals with detectable HBV DNA?

Response: Thank you very much for this response. We were able to detect HBV/HDV co-infection in 5 of the 51 HBsAg positives published in reference 27 

Round 2

Reviewer 3 Report

Comments and Suggestions for Authors

No further comment. It could be accepted in present form

Comments on the Quality of English Language

No further comment. It could be accepted in present form